# Synthesis of γ-Aminobutyric Acid-Modified Chitooligosaccharide Derivative and Enhancing Salt Resistance of Wheat Seedlings

**DOI:** 10.3390/molecules27103068

**Published:** 2022-05-10

**Authors:** Wenyun Wang, Song Liu, Mingyan Yan

**Affiliations:** 1College of Chemistrye and Molecular Engineering, Qingdao University of Science and Technology, Qingdao 266042, China; wangwenyunqust@163.com; 2CAS and Shandong Province Key Laboratory of Experimental Marine Biology, Center for Ocean Mega-Science, Institute of Oceanology, Chinese Academy of Sciences, Qingdao 266071, China; 3University of Chinese Academy of Sciences, Beijing 100049, China; 4Laboratory for Marine Drugs and Bioproducts, Pilot National Laboratory for Marine Science and Technology (Qingdao), No. 1 Wenhai Road, Qingdao 266237, China

**Keywords:** chitooligosaccharide, γ-aminobutyric, salt stress

## Abstract

Salinity is one of the major abiotic stresses limiting crop growth and productivity worldwide. Salt stress during germination degenerates crop establishment and declines yield in wheat, therefore alleviating the damage of salt stress to wheat seedlings is crucial. Chitooligosaccharide (COS) was grafted with γ-aminobutyric acid based on the idea of bioactive molecular splicing, and the differences in salt resistance before and after grafting were compared. The expected derivative was successfully synthesized and exhibited better salt resistance-inducing activity than the raw materials. By activating antioxidant enzymes such as superoxide dismutases (SOD), catalase (CAT) and phenylalanine ammonia-lyase (PAL) and subsequently eliminating reactive oxygen species (ROS) in a timely manner, the rate of O^−^_2_ production and H_2_O_2_ content of wheat seedlings were reduced, and the dynamic balance of free radical metabolism in the plant body was maintained. A significantly reduced MDA content, reduced relative permeability of the cell membrane, and decreased degree of damage to the cell membrane were observed. A significant increase in the content of soluble sugar, maintenance of osmotic regulation and the stability of the cell membrane structure, effective reduction in the salt stress-induced damage to wheat, and the induction of wheat seedling growth were also observed, thereby improving the salt tolerance of wheat seedlings.

## 1. Introduction

Soil salinization is one of the major constraints for agriculture in semiarid ecosystems and threatens food security on a global scale, with affected areas spreading annually and predicted to reach 50% of arable land by 2050 [1]. Therefore, alleviating the damage to plants caused by salt stress and improving crop yield have become the focus of many researchers. Wheat (*Triticum aestivum* L.) is a popular staple food of more than one-third of the global population, and wheat grain yield losses are increasing up to 60% due to the adverse effects of soil salinity on approximately 20% of the agricultural land worldwide [2,3]. In addition, agricultural soils in arid regions accumulate higher amounts of salts after water evaporation in the summer and subsequent cultivation of wheat in the winter, thus causing more severe damage at the germination and seedling stages than at the developmental phases [4]. Increasing its wheat production in normal and stressed environments is of utmost importance. With continuous research on plant resistance, methods to improve the salt tolerance of crops include the selection and breeding of salt-tolerant crop varieties, genetic engineering to modify crops, and the application of exogenous inducers. Currently, the application of exogenous substances to improve salt tolerance in plants has become a hot topic in salt resistance research. Among these exogenous substances, chitosan (CTS) and its derivatives deserve particular attention, because they have been shown to counteract the detrimental effects of both biotic and abiotic stresses and increase the yield and quality of crops. Chitosan (CTS), an alkaline cationic polysaccharide formed by the deacetylation of chitin, is a natural, low-toxicity, biodegradable, environmentally friendly, renewable, and inexpensive resource and has many applications in the agriculture sector. Since the discovery of CTS by Rouget in 1859 [5], several studies have indicated its role in enhancing plant growth and increasing the abiotic stress tolerance of plants, including rice, maize, safflower, sunflower, and creeping bentgrass [6,7,8,9,10]. Upon NaCl treatment, low-medium molecular weight chitosan alleviates the inhibitory effect of salt stress on the biomass of durum wheat [11]; chitosan pretreatment of creeping grass seeds significantly reduces the Na^+^ content of plant roots, which regulates the salt oversensitive pathway and upregulated the expression of salt tolerance genes [12]. Chitosan alleviates plant injury caused by abiotic stresses, such as low temperature and drought, by participating in the regulation of plant osmoregulatory substances and increasing the activity of antioxidant enzymes [13,14], suggesting that it plays an important role as a novel anti-stress agent in agricultural production. Currently, chitosan and its derivatives are increasingly used to resist plant stress. γ-Aminobutyric acid (GABA) is a nonprotein amino acid that is widely distributed throughout plants and humans. GABA probably plays a dual role in plants as both a signaling molecule and a metabolite. Moreover, GABA might be involved in the regulation of physiological and biochemical reactions in plant cells, such as growth and development, signal transmission, and carbon and nitrogen balance. According to a previous study, GABA itself exhibits ROS scavenging ability [15]. GABA treatment of maize seedlings under salt stress effectively reduces oxidative damage, reduces damage to the photosynthetic system, improves photosynthesis and chlorophy Ⅱ fluorescence parameters, and enhances the salt tolerance of seedlings [16]. Exogenous GABA significantly improves the heat tolerance of creeping bentgrass and affects hormone signaling pathways in woody plants under salt stress conditions [17,18]. A GABA soaking treatment of white clover seeds [19] before sowing is beneficial to increase the endogenous GABA content in plants, promoting starch catabolism metabolism, antioxidant enzyme activities, and gene transcription levels in seedlings, as well as mitigating the damage of salt stress.

Appropriate chemical structural modifications of chitosan molecules have been shown to significantly increase its activity and enable a wider range of applications. For example, COS-N-Ger prepared from chitosan and geraniol significantly inhibits the growth of *Escherichia coli* and *Staphylococcus aureus* [20]. The sulfated chitooligosaccharide (SCOS) prepared by Zou et al. [21] was used to treat wheat seedlings under salt stress and reduced the MDA content in plants and increased the photochemical efficiency and fluorescence parameters. The exogenous application of both chitosan and γ-aminobutyric acid induces salt resistance in plants. Therefore, a novel chitooligosaccharide-γ-aminobutyric acid derivative (COS-G) was prepared by coupling chitooligosaccharides and γ-aminobutyric acid through the principle of active splicing, and its effect on inducing plant salt resistance was evaluated.

## 2. Results

### 2.1. Characterization of Chitooligosaccharide Derivatives

#### 2.1.1. FT-IR Spectroscopy

The FT-IR spectra of COS and COS-G are shown in Figure 1. The spectrum of raw COS (1 K) displays the C = O stretching vibration absorption peak and the N–H bending vibration absorption peak at 1645 cm^−1^ and 1573 cm^−1^, respectively [22]. This indicates that the –NH_2_ stretching vibration peak of chitosan at 1573 cm^−1^ disappears after its modification by γ-aminobutyric acid, and two new absorption peaks appeared at 1640 cm^−1^ and 1535 cm^−1^. These two absorption peaks are attributed to amide I generation and the NH deformation vibration absorption peak of amide II generation [23], indicating that an amide bond is formed and the reaction occurs on the 2–NH_2_ group, which proves that γ-aminobutyric acid was successfully grafted to chitooligosaccharide.

#### 2.1.2. NMR Spectroscopy

The NMR hydrogen spectrum (^1^H-NMR) of COS is shown in Figure 2. According to the literature [24], 3.34–3.8 ppm are the proton peaks of H3, H4, H5, and H6, respectively, 4.78 ppm is the proton peak of H1, 2.93 ppm is the hydrogen proton peak on 2–NH_2_, and 1.779 ppm is the methyl peak of the COS backbone –CH3 [25]. Figure 3 shows the spectrum of COS-G. In addition to maintaining the proton peaks of chitosoligosaccharide, proton peaks at three positions: 3.92 ppm, 2.29 ppm and 1.61 ppm, have been added. After consulting the literature, these three hydrogen absorption peaks are attributed to positions 9, 7, and 8 of γ-aminobutyric acid, respectively. Based on this result, γ-aminobutyric acid successfully reacted with chitooligosaccharide to generate a derivative (COS-G).

#### 2.1.3. The Degree of Substitution of the Chitooligosaccharide Derivative

The substitution degree calculated from the NMR hydrogen spectra of the derivatives is 63.8%.

### 2.2. The Effect of COS-G on Wheat Seedlings under Salt Stress

#### 2.2.1. Plant Biomass Accumulation

Salt stress significantly inhibited the growth of the wheat shoot length, which was 35.3% lower than that of the CK group. Treatment with exogenous substances alleviated the growth inhibition of seedlings under salt stress, and Sodium nitroprusside (SNP), GABA, COS, and COS-G significantly increased growth by 21.2%, 17.9%, 20.2%, and 24.1%, respectively, compared with the NaCl-treated group (*p* > 0.05). Among them, COS-G exerted better effects on alleviating salt stress than the other treatments.

Salt stress significantly inhibited the root growth of wheat seedlings, and the root length of wheat seedlings was 56.8% lower than that of the CK group, while the application of exogenous substances alleviated the inhibitory effect of salt stress on roots. The SNP, GABA, COS, and COS-G treatments increased root growth by 10.7%, 14.2%, 0.8%, and 24.1%, respectively, compared with the NaCl group, with COS-G exerting a significant effect on increasing root growth.

Salt stress led to a significant decrease in the shoot fresh weight compared with the CK group, and the shoot fresh weight in the salt stress group decreased by 57.2% compared with the CK group. The exogenous application of SNP, GABA, COS, and COS-G increased the fresh leaf weight by 23.1%, 24.7%, 22.6%, and 48.9%, respectively, compared with the NaCl group, and the best effect was observed in the COS-G group. The application of exogenous substances alleviated the inhibitory effect of salt stress on the growth of wheat seedlings to a certain extent, thus promoting an increase in biomass.

Due to salt stress, the wheat shoot fresh weight decreased. Thus, the decrease in the dry matter accumulation, as evidenced by the determination of wheat shoot dry weight, revealed that the wheat shoot dry weight of the salt stress treatment group was reduced by 47.9% compared with the CK group, as shown in Figure 3d. Treatment with exogenous substances did not result in a significant difference between the groups exposed to NaCl but increased the shoot dry weight to some extent. The effects of GABA and COS-G on increasing dry weight were better than those of SNP and COS and increased by 20.4% and 28.2% compared with the NaCl group, respectively. Based on these results, COS-G increased the accumulation of organic matter in plants to a certain extent and improved the ability of wheat seedlings to resist salt stress.

#### 2.2.2. Soluble Sugar and Soluble Protein Contents

Soluble sugars and soluble proteins are important osmoregulatory substances in plants [26] that regulate the osmotic potential of plants to resolve salt stress. As shown in Figure 4, the soluble sugar content was reduced by 36.8% (*p* < 0.05) in wheat under salt stress compared to the CK group. After treatment with different exogenous substances, the soluble sugar content increased significantly in each treatment group. The GABA and COS-G groups showed a significantly higher soluble sugar content than the salt stress group by 216.7% and 342.4%, respectively. The experimental results indicated that the effect of COS-G on increasing the soluble sugar content in wheat leaves was significant compared with that of the other treatment groups and effectively regulated the cellular osmotic pressure. The soluble protein content increased by 3.0% (*p* < 0.05) under salt stress compared with the CK group. COS-G promoted the accumulation of osmoregulatory substances in wheat seedlings under salt stress and helped alleviate the stress of wheat seedlings. The effect of COS-G on the soluble sugar contents of wheat seedlings was greater than that on the soluble protein contents.

#### 2.2.3. MDA Contents

Lipid peroxidation occurs in plants under salt stress, and a large number of free radicals cause cell damage [27]. As one of the products of lipid peroxidation, the MDA content is proportional to the degree of stress. The effects of SNP, GABA, COS, and COS-G on the MDA content in seedlings under salt stress are shown in Figure 5. The MDA content in wheat seedlings under salt stress was 2.32 times higher than that in the CK group (*p* < 0.05). Meanwhile, in the groups treated with SNP, GABA, COS, and COS-G, the MDA content in the leaves of wheat seedlings was significantly reduced. The MDA content was 18.5%, 30.5%, 35.0%, and 43.0% lower than that of the salt stress group, respectively, with all reaching significant levels (*p* < 0.05), with the greatest reduction observed in the COS-G group. Thus, COS-G effectively reduced the MDA content in plants and alleviated the oxidative damage to wheat seedlings induced by salt stress.

#### 2.2.4. O^−^_2_ Production Rate and H_2_O_2_ Contents

Under salt stress, plants produce large amounts of reactive oxygen species (ROS), mainly in the form of O^−^_2_ and H_2_O_2_ [28]. The excess activity of these molecules damages intracellular proteins, unsaturated fatty acids, and other macromolecules, affecting the normal function of plant cell membranes and subsequently inhibiting plant growth and development [29]. Wheat seedlings produced large amounts of O^−^_2_ under salt stress, as shown in Figure 6a. The NaCl treatment increased the rate of O^−^_2_ production in wheat seedlings by 11.0% compared with the CK group. Compared with the salt stress group, the rate of O^−^_2_ production in wheat seedlings after treatment with different exogenous substances showed a decreasing trend, among which the COS-G group showed a more obvious decrease of 12.5% (*p* < 0.05), indicating that COS-G inhibited the rate of O^−^_2_ production in plants under salt stress to some extent, and the O^−^_2_ content was restored to the normal level.

The NaCl treatment resulted in a highly significant increase in the H_2_O_2_ content in wheat seedlings compared to the group without salt stress treatment (CK), as shown in Figure 6b, which was 43.2% higher than the CK group. The accumulation of H_2_O_2_ induced by salt stress was alleviated by the treatment with different exogenous substances, and the SNP, GABA, COS, and COS-G groups showed decreases of 23.5%, 15.9%, 40.8%, and 71.8%, respectively, compared with the NaCl group, with the most significant effect observed for COS-G (*p* < 0.05). The results showed that COS-G effectively scavenged excessive ROS in plants and alleviated the salt stress-induced damage to wheat.

#### 2.2.5. Antioxidant Enzyme Activities

Antioxidant enzyme activity is an important indicator of abiotic stress, and the main antioxidant enzymes expressed in plants are superoxide dismutase (SOD), peroxidase (POD), catalase (CAT) and phenylalanine aminolase (PAL) [30]. As shown in Figure 7, SOD, POD, and PAL levels in plants under salt stress were 7.4%, 59.6%, and 26.9% higher than those in the CK group, respectively. A significant difference in the change in SOD activity was not observed between plants under salt stress and the CK group (*p* > 0.05). CAT activity showed an opposite trend in plants under salt stress, decreasing by 26.7% compared to the blank group. After treatment with different exogenous substances, SOD activity decreased in the SNP, GABA, COS, and COS-G groups and increased by 30.5%, 6.1%, and 19.6%, respectively, compared with the CK group. The effect of GABA was better than that of COS and COS-G, and the difference was significant (*p* > 0.05). The POD activity was significantly increased in plants under salt stress by 61.5% (*p* < 0.05) compared to the CK group, and the enzyme activity was reduced by SNP treatment, while the levels observed after GABA, COS, and COS-G treatment were increased by 29.0%, 27.0%, and 13.9%, respectively, compared with the CK group. However, these values were significantly lower than those in the salt stress group (*p* < 0.05). CAT activity decreased under salt stress, and the application of SNP and COS-G effectively increased CAT enzyme activity, which was 64.7% and 45.9% higher than that in the salt stress group, respectively. These effects were better than those of the GABA and COS treatments and significantly different from that in the salt stress group. PAL activity was increased by different treatments compared to the salt stress group, with COS and COS-G increasing its activity by 31.6% and 21.2%, respectively.

#### 2.2.6. Total Antioxidant Capacity (T-AOC)

When the plant is experiencing adverse stress, various antioxidant enzymes and antioxidant substances are produced by different sources and in different quantities to resist the damage of oxygen free radicals generated in response to stress [31]. Since the antioxidant capacity of individual molecules is difficult to measure and interactions among antioxidant enzymes and antioxidants have been identified, a more meaningful approach is to determine the comprehensive total antioxidant capacity rather than the contents of specific antioxidants. The effects of different exogenous substances on the total antioxidant capacity (T-AOC) of wheat seedlings under salt stress are shown in Figure 8. Compared with the CK group, the NaCl group decreased the total antioxidant capacity by 68.7%, and compared with the NaCl group, the total antioxidant capacity of the SNP and GABA treatment groups did not change significantly. Meanwhile, COS and COS-G significantly increased the total antioxidant capacity of wheat seedlings by 464% and 521% (*p* < 0.05), respectively, compared with the NaCl group, indicating that COS-G induced and activated the antioxidant system of wheat seedlings, scavenged excess free radicals, regulated the antioxidant capacity of plants, and alleviated the oxidative damage to plants caused by salt stress.

## 3. Discussion

Salt stress exerts many adverse effects on plants, resulting in significant changes in growth conditions, morphological characteristics, and physiological indicators. Wheat seeds at the germination stage are most sensitive to salt and display a significantly reduced biomass under salt stress. Odat et al. [32] showed that embryonic axis length and the root dry weight of wild peas were reduced by different salinity levels and that chitosan treatment exerted a mitigating effect on these changes. Reports indicated that foliar spraying of chitosan inducer improved the trait characteristics of maize [33] and rice [34] and improved plant tolerance to stress conditions. As shown in the present study, salt treatment reduced the biomass indices, such as shoot length, root length, shoot fresh weight, shoot dry weight, and of wheat seedlings by 35.3%, 56.8%, 57.2%, 28.2%, respectively, compared with the NaCl treatment, with the most pronounced effect observed on reducing fresh weight. The exogenous application of COS-G alleviated the inhibitory effect of salt stress on wheat shoot length and root length, and the shoot fresh weight of seedlings showed an increasing trend of 48.9%, which was better than that of the raw materials and SNP. It also exerted a positive effect on maintaining normal plant growth.

The large amount of Na^+^ in the salt environment caused osmotic dysregulation in plant cells, and the plants enhanced their adaptability to NaCl stress by accumulating small-molecule organic substances such as soluble sugars and soluble proteins to balance the intracellular water potential and maintain cellular osmotic pressure homeostasis. Elsamad et al. [35] found that the salt tolerance of Clark and Forrest varieties of soybean is associated with the accumulation of soluble protein and proline, while salt tolerance in the Kint soybean variety is associated with reduced soluble sugar and soluble protein contents. According to the literature, exogenous applications of substances such as SA [36], GABA [37], and melatonin [38] improve the osmoregulatory capacity of plants and protect the activity of intracellular enzymes required to maintain cellular activity. In the present study, the soluble sugar content was significantly reduced in plants under salt stress, and exogenous COS-G effectively increased the soluble sugar and soluble protein contents in wheat seedlings. COS promoted the accumulation of soluble proteins, thus showing the inconsistent effects of different substances on soluble sugars and soluble proteins. COS-G exerted the most significant effect on the accumulation of soluble sugars, the levels of which increased by 342.4% compared with plants under salt stress to promote the C and N metabolism in wheat seedlings and accelerate the synthesis of organic matter, thus enhancing the salt tolerance of wheat seedlings.

Plants have developed a series of protective mechanisms to adapt to different stressful environments during their long-term evolution, among which the scavenging of ROS by antioxidant enzyme systems is one of the important methods to maintain normal plant growth under salt stress [39]. When plants are under salt stress, an ROS imbalance is the initial response of plants to adversity. At this time, the rate of ROS production is greater than the rate of scavenging, which will cause oxidative damage to plants. MDA is one of the main products of lipid peroxidation and is an important indicator of the degree of membrane lipid peroxidation and plasma membrane damage in plant cells [40]. Here, the MDA content in plants under salt stress was significantly increased, indicating that salt stress caused a substantial accumulation of free radicals in cells, and the exogenous application of COS-G reduced the malondialdehyde content in plants and alleviated the damage to plants induced by membrane lipid peroxidation. The increase in intracellular excess reactive oxygen species content also induced a defense response of the intracellular reactive oxygen species scavenging system in plants, and the antioxidant enzyme system, including superoxide dismutase, catalase, and peroxidase, is one of the important defense systems of plants in response to the adverse stress [41].

A large amount of ROS is generated in plants under salt stress, leading to membrane lipid peroxidation reactions, destabilization of cell membranes, reduced antioxidant enzyme activities, and slowed metabolism [42]. Zou et al. [21] prepared sulfated chitooligosaccharide (SCOS) from chitosan as a raw material, and the application of exogenous SCOS increased the chlorophy content and decreased the MDA content in wheat, along with improving the salt resistance of wheat by regulating antioxidant enzyme activity. SCOS also regulated mRNA expression levels in plants, which effectively alleviates the stress-induced damage to wheat seedlings. In the present study, SOD and POD activities increased in plants under salt stress, indicating that harmful substances such as ROS stimulated the production of related enzymes and increased enzyme activity, and the application of COS-G effectively increased SOD activity, while POD activity showed a decreasing trend after chitosan derivative treatment. The explanation for this phenomenon may be that with increasing treatment time, intracellular H_2_O_2_ content gradually decreased, and enzyme activity subsequently decreased and tended to return to the normal value of the blank group. Salt stress inhibited CAT activity, and the CAT content increased by 45.9% after the COS-G treatment compared with the salt stress group, enabling the CAT content to return to a near normal level. When plants were subjected to abiotic stress, COS-G increased PAL activity and the T-AOC by up to 21.2% and 521%, respectively, which enhanced the tolerance of plants to abiotic stress.

COS-G exerts varying effects on increasing SOD, POD and CAT activities and exerts a significant effect on increasing the T-AOC. In general, the ability to scavenge ROS was better than that of the other groups, indicating that COS-G enables plants to withstand salt stress. The synergistic effects of antioxidant enzymes in wheat seedlings reduced the rate of ROS production and enhanced the salt tolerance of wheat seedlings.

## 4. Materials and Methods

### 4.1. Materials

Chitooligosaccharide (Mw = 1000 Da) was purchased form Yunzhou Biological Technology Co. Ltd, Qingdao, China, Chemical grade; γ-Aminobutyric acid was purchased form Yuanye Biological Technology Co. Ltd, Shanghai, China, analytical grade; 1-Ethyl-(3-dimethylaminopropyl) carbodiimide hydrochloride (EDC·HCl) was purchased form Solarbio, Beijing, China, analytical grade; N-Hydroxysuccinimide (NHS) was purchased form Yuanye Biological Technology Co. Ltd., Shanghai, China, analytical grade; Morpholine ethyl sulfonic acid (MES) was purchased form Aladdin, Shanghai, China, analytical grade; dialysis bags MWCO = 500–1000 Da was purchased form Spectral Medicine, China; a PAL assay kit and T-Aoc assay kit was purchased form Solarbio, Beijing, China.

### 4.2. Preparation of the Chitooligosaccharide Derivative (COS-G)

The method was suitably improved according to a previous study [43]. We first dissolved 0.7218 g γ-aminobutyric acid, 4.025 g EDC and 2.416 g NHS in 50 mL 0.1 mol/L MES buffer. The pH was adjusted to 5.5 with NaOH solution to fully dissolve the reactants in the system and form a homogeneous reaction system, and the carboxyl groups of γ-aminobutyric acid were activated by stirring at room temperature for 3 h. Next, 1000 Da chitooligosaccharides were added to the reaction system, and the reaction was incubated at room temperature for 24 h. After the reaction was completed, it was put into a dialysis bag with a molecular weight cut-off of 100–500Da, dialyzed in distilled water for five days, and the distilled water was replaced every 6h to ensure that the reactants that do not participate in the reaction are removed, and then the dialysate was freeze-dried to obtain chitooligosaccharide aminobutyric acid derivative (COS-G).

COS-G was further characterized using FT-IR and NMR spectroscopy. Fourier transform infrared (FT-IR) spectra of samples were detected in the range of 4000 to 400 cm^−1^ using an FT-IR spectrometer (Thermo Scientific Nicolet iS10, Waltham, MA, USA). ^1^H NMR and ^13^C NMR spectra were recorded using an NMR spectrometer (JEOL JNM-ECP600, Japan). ^1^HNMR and ^13^CNMR spectra were recorded on a 500 MHz Agilent DD2 500 spectrometer for structural analysis in COS and COS-G. Measurements were recorded at 25 °C. Samples were dissolved at D_2_O (10 mg/mL), and the pulse sequence was s2pul and the relaxation delay was 1.0000 s.

### 4.3. Effect of COS-G on Wheat Seedlings under Salt Stress

#### 4.3.1. Seed Dipping Treatment

The wheat (Changfeng, 2112) seeds were sterilized with 75% ethanol for 30 s, rinsed repeatedly with deionized water, and subsequently soaked in deionized water, SNP, GABA, 1 K COS, and COS-G for 8 h at room temperature. Six treatment groups were established: ① CK, deionized water; ② NaCl, deionized water; ③ SNP, 0.08 mmol/L NO solution for soaking; ④ GABA, 500 mmol/L GABA solution for soaking; ⑤ 1K COS, 500 mmol/L 1000 Da COS for soaking; ⑥COS-G, 500 mmol/L COS-G solution for soaking. The results of the selection are listed in Table 1.

#### 4.3.2. Seedling Development

After the soaking solution was poured off, full wheat seeds with a uniform size were selected and placed in Petri dishes with plastic nets, and a 100 mmol/L NaCl solution was added to half of the wheat seeds. Three replicates were germinated in an incubator at 25 °C for 24 h in the dark, during which water was continuously replenished. After three days, seeds were transferred into a Hoagland nutrient solution containing 100 mmol/L NaCl. The following growth conditions were used: a photoperiod of 14 h/10 h and a temperature of 25 °C/19 °C. Wheat was foliar sprayed with a 100 mmol/L solution of the corresponding treatment daily after the appearance of the first leaf. Fifteen seedlings were randomly selected after 15 days to determine shoot length, root length, seedling fresh weight and other biomass indicators.

### 4.4. Determination of Relevant Physiological Indicators of Seedlings

After wheat seedlings were treated with different compounds for 15 days, the inverted second leaves of wheat from different treatment groups were cut, ground into a powder with liquid nitrogen, and stored in an ultralow temperature freezer at −80 °C.

#### 4.4.1. Determination of the MDA Content

Plant leaves were extracted with TBA in boiling water for 15 min, and the absorbance was measured at 450 nm, 532 nm and 600 nm.

#### 4.4.2. Determination of the ROS

The rate of O^−^_2_ production was determined using the method reported by Tuba et al. [44], and the absorbance of the extract was measured at 540 nm. The H_2_O_2_ level was determined using the method reported by Velikova et al. [45], and the absorbance was measured at 240 nm.

#### 4.4.3. Determination of the Osmoregulatory Substance Content

The soluble sugar content was determined using the anthrone colorimetric method by measuring the absorbance at 620 nm [46], and the soluble protein content was determined using the Coomassie Brilliant Blue dye method by measuring the absorbance at 595 nm [47].

#### 4.4.4. Determination of Antioxidant Enzyme Activity

SOD activity was determined using the nitrogen blue tetrazolium reduction method [48], 0.1 g of fresh tissue was collected, and absorbance values were measured at 560 nm using the nitrogen blue tetrazolium reduction method. POD activity was determined using the guaiacol method [49], and changes in absorbance were measured at 470 nm for one minute. Changes in CAT activity were determined using the method described by Kochba et al. [50] by monitoring the absorbance at 240 nm for one minute. PAL activity was determined using the phenylalanine aminolase enzyme activity kit (Solarbio, Beijing, China).

#### 4.4.5. Determination of the Total Antioxidant Enzyme Capacity (T-AOC)

T-AOC activity was determined using the Total Antioxidant Capacity kit (Solarbio, Beijing, China).

### 4.5. Statistical Analysis

Each test was performed in triplicate, and the results were averaged. Origin 8.5 software was used to process the data, and data were subjected to ANOVA analysis by SPSS (version 19.0) and Duncan’s test (*p* < 0.05) to compare the mean value of different treatments. Each of the data points was expressed as the average ± SD of three independent replicates. (*p* < 0.05).

## 5. Conclusions

Using the principle of active group splicing, γ-aminobutyric acid and chitooligosaccharide were organically synthesized into one molecule, the chitooligosaccharide γ-aminobutyric acid derivative was successfully prepared, and its effect on the salt resistance of wheat seedlings was studied. The chitooligosaccharide γ-aminobutyric acid derivative increased the antioxidant enzyme activity in wheat seedlings, effectively scavenged oxygen radicals, significantly increased the antioxidant capacity of plants, substantially reduced the MDA content in the leaves of wheat seedlings, reduced the degree of cell membrane damage, increased the soluble sugar content to accelerate plant metabolism, and provided energy to promote plant growth and development under salt stress. The mechanism of salt tolerance in plants is a complex process, and the mechanism of action of chitooligosaccharide derivatives requires further investigation.

## Figures and Tables

**Figure 1 molecules-27-03068-f001:**
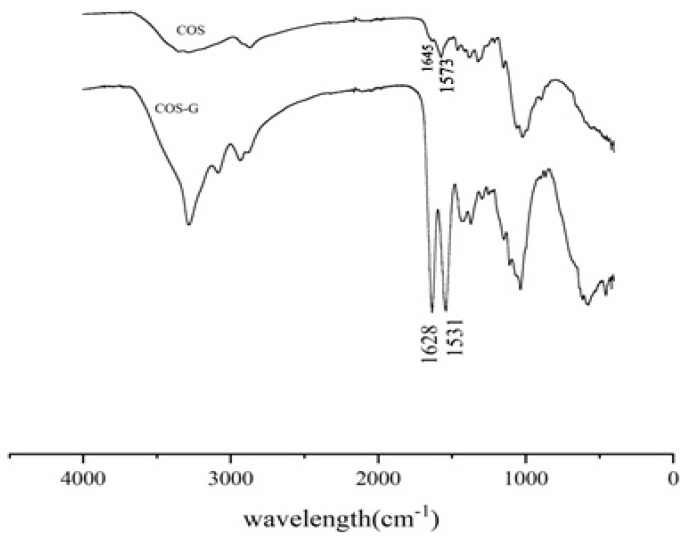
Infrared spectra of COS and COS-G.

**Figure 2 molecules-27-03068-f002:**
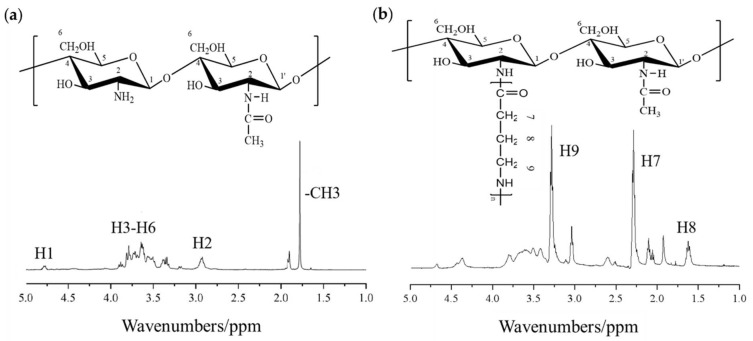
Nuclear magnetic resonance hydrogen spectra of COS (**a**) and COS-G (**b**).

**Figure 3 molecules-27-03068-f003:**
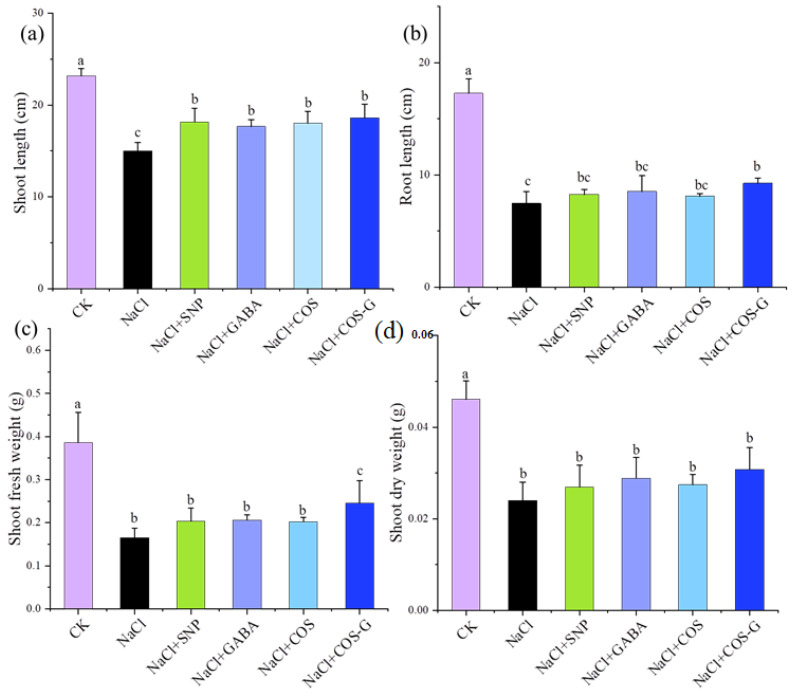
Effects of different treatment groups on wheat seedling biomass under salt stress. (**a**) shoot length; (**b**) root length; (**c**) shoot fresh weight; (**d**) shoot dry weight. Values are the mean ± SD of three replicates in independent experiments. Different letters (^a,b,c^) indicate significant differences at *p* < 0.05 on the same parameter.

**Figure 4 molecules-27-03068-f004:**
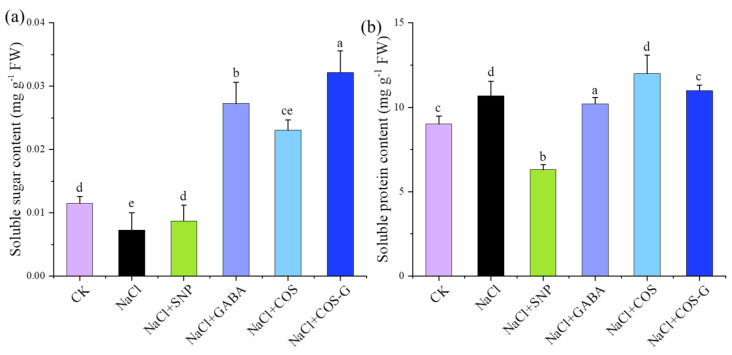
Effects of different treatment groups on osmotic substances in wheat seedling leaves under salt stress. (**a**) soluble sugar content; (**b**) soluble protein content. Values are the mean ± SD of three replicates in independent experiments. Different letters (^a,b,c,d,e^) indicate significant differences at *p* < 0.05 on the same parameter.

**Figure 5 molecules-27-03068-f005:**
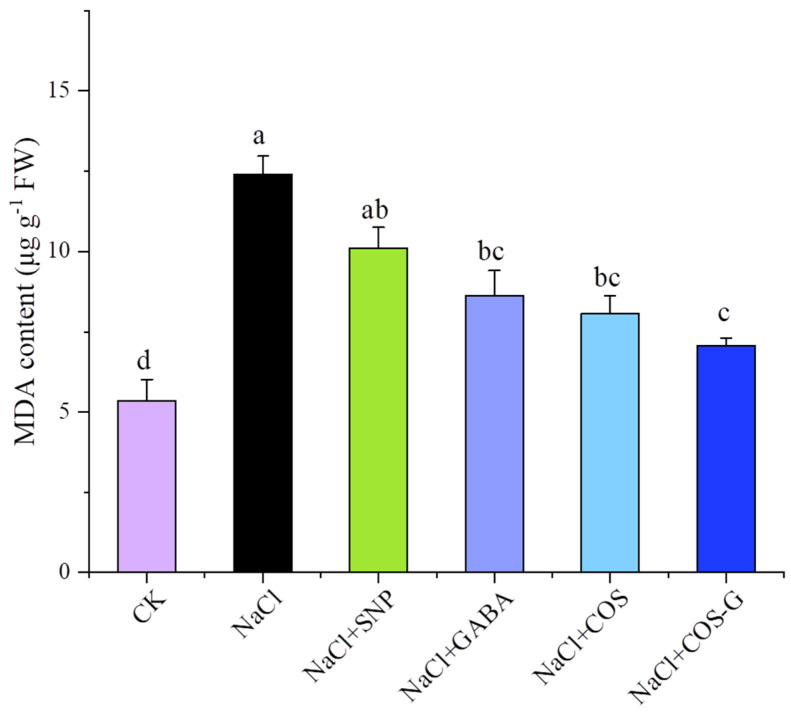
Effect of different treatment groups on malondialdehyde (MDA) content of wheat seedling leaves under salt stress. Values are the mean ± SD of three replicates in independent experiments. Different letters (^a,b,c,d^) indicate significant differences at *p* < 0.05 on the same parameter.

**Figure 6 molecules-27-03068-f006:**
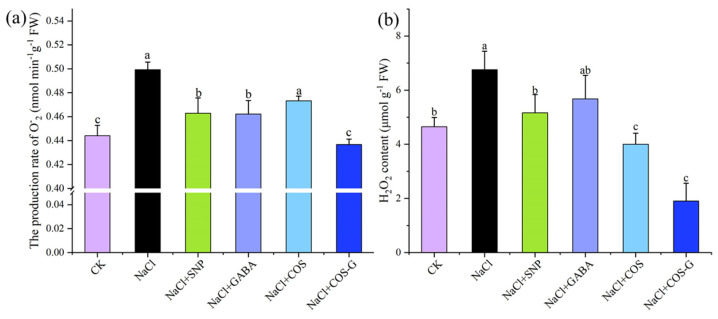
Effects of different treatment groups on O^−^_2_ and H_2_O_2_ content in wheat seedling leaves under salt stress. (**a**) the production rate of O^−^_2_; (**b**) H_2_O_2_ content. Values are the mean ± SD of three replicates in independent experiments. Different letters (^a,b,c^) indicate significant differences at *p* < 0.05 on the same parameter.

**Figure 7 molecules-27-03068-f007:**
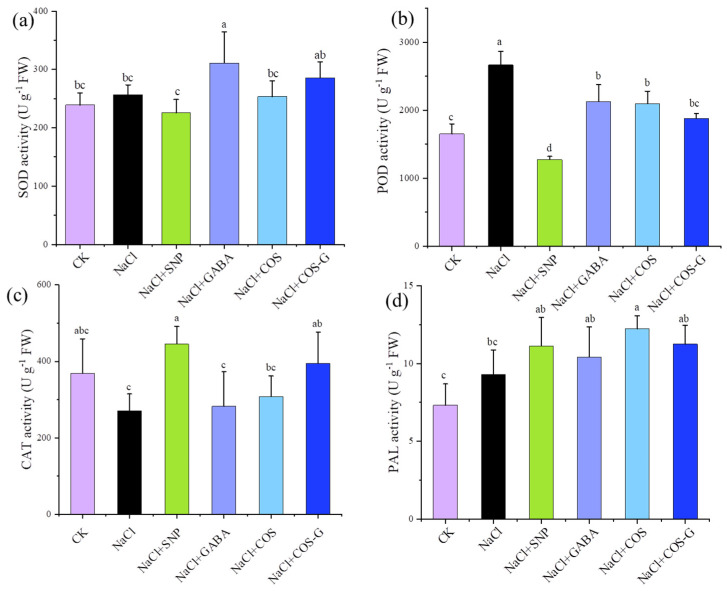
Effects of different treatment groups on the activity of antioxidant enzymes in wheat seedling leaves under salt stress. (**a**) superoxide dismutase (SOD); (**b**) peroxidase (POD); (**c**) catalase (CAT); (**d**) phenylalanine aminolysis enzyme (PAL). Values are the mean ± SD of three replicates in independent experiments. Different letters (^a,b,c,d^) indicate significant differences at *p* < 0.05 on the same parameter.

**Figure 8 molecules-27-03068-f008:**
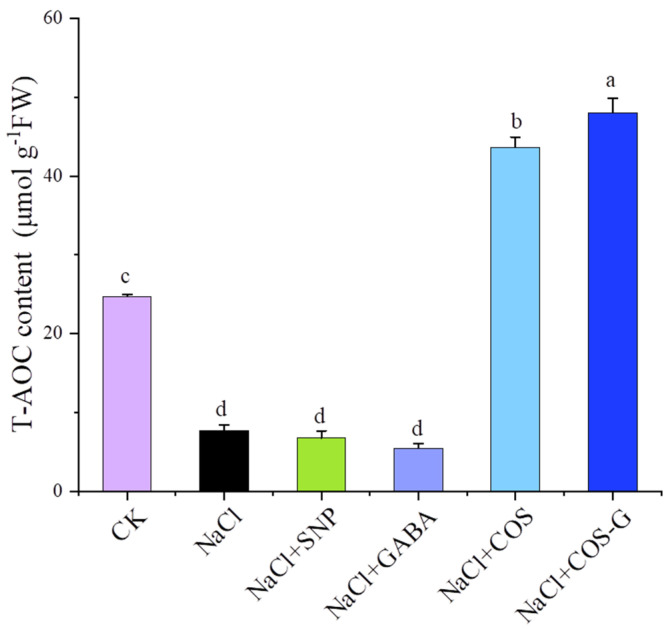
Effects of different treatment groups on T-AOC in wheat seedling leaves under salt stress. Values are the mean ± SD of three replicates in independent experiments. Different letters (^a,b,c,d^) indicate significant differences at *p* < 0.05 on the same parameter.

**Table 1 molecules-27-03068-t001:** Different treatment combinations for wheat seeds.

Treatment	NaCl (mmol/L)	Concentration
CK	0	0
NaCl	100	0
SNP	100	0.08 mmol/L
GABA	100	500 mg/L
COS	100	500 mg/L
COS-G	100	500 mg/L

## Data Availability

Not applicable.

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
