# Peer review of "Synthesis of γ-Aminobutyric Acid-Modified Chitooligosaccharide Derivative and Enhancing Salt Resistance of Wheat Seedlings"

_molecules, 2022, doi:10.3390/molecules27103068_

Round 1

Reviewer 1 Report

In this study, Wang et al. synthesized a γ-aminobutyric acid-modified chitooligosaccharide derivative which was further tested for enhancing salt resistance of wheat seedlings. In my opinion, the study is planned nicely, and the findings are interesting. The manuscript is also written well. However, I have few comments that can be considered for improving the quality of manuscript.

  • Whether authors tested the effect of COS-G on wheat seedlings under control conditions i.e. without any stress. It will be good to include that data to confirm COS-G role in stress tolerance.
  • Please provide a picture of wheat seedlings under different conditions to show the effect of COS-G on plants.
  • Fig 3a: What do you mean by seedlings length? Is this length of whole seedling or just shoot? If it is shoot, then change the name of axis from wheat length to shoot length. Please also change this throughout manuscript.
  • Please expand the legends of all figures and mention the replicates and p-value information in the legends.
  • What is SNP? Please provide full name of this. Abbreviations must be defined at their first mention and used accordingly.
  • Methodology section needs improvement, for example how each experiment was conducted and analyzed. Further, the detail of chemicals and kits i.e. company name, location and catalogue number should be included.
  • Abstract: Please provide some background information.
  • There are several typographical, and errors throughout the manuscript. Authors need to check that.

Reviewer 2 Report

The research article: "Synthesis of γ-aminobutyric acid modified chitooligosaccharide derivative and increasing salt resistance of wheat seedlings" is well thought out and carefully prepared. I found only a few minor errors, mostly editorial.

  1. I did not find any information on how many repetitions the individual experiments were carried out.
  2. There is no data on the NMR spectrometer settings.
  3. The Authors should pay attention to the direction of the ”<” mark in the significance level (P < 0.05). In a few cases it is opposite (P > 0.05). P < 0.05 was implemented with or without spaces in the text. The notations should be standarized. Check on lines: 160, 177, 212, 218, 219, 222.
  4. In Materials and Methods, the Authors used the Bradford method (and coomassie brilliant blue dye) to measure the protein content, and not, as they wrote: ”The Komas Brilliant Blue method”.
  5. When using the kit, please provide its symbol and the company from which it was obtained (lines 392 and 394).
  6. Line 34: The abbreviation for Linnaeus should not be written in italics.
  7. Line 69: The word chlorophyll is written using Roman numerals instead of the letters.
  8. Line 79: The Latin names of two bacteria should be written in italics.
  9. Line 79: The article reference number is missing from the citation. This problem also appears on lines 254, 270, 302. Check the entire manuscript against this.
  10. In chapter 2.1.1. pay attention to the writing of single bonds. They should be represented by a dash (–) instead of a hyphen (-) (lines 90 and 91).
  11. The superoxide is usually written with a minus next to the oxygen atom rather than the number two. The Authors’ notation may be misleading.
  12. The Authors should pay attention to the description of the Y axis in Figures 3, 4, 6, 7 and 8. They are not thorough. There are unnecessary symbols ”/” in front of the unit, missing spaces or additional spaces. Also, the FW notation is not uniform (twice Fw).
  13. The entire text should be checked for missing or extra spaces and capitalized words inside the sentences.

April 28th 2022

Reviewer 3 Report

The manuscript is dealing with improvement of plant tolerance to salt by application of chitooligosaccharide derivative.

The manuscript is well written, however the following items should be corrected:

1) Latin names of bacteria when used for the first time should be given in full forma and should be written using Italic font.

2) Each abbreviation when used for the first time should be explain, e.g. line: 79.

3) Authors should follow the rules when preparing this text e.g. Fig. 1 which is incorrect.

4) What means letters above columns? Statistical significance?

5) Figure 8 - this caption is unclear, please improve.

Round 2

Reviewer 1 Report

Authors have responded to my comments satisfactorily therefore manuscript may be accepted for publication in the current form.